# Differential Metabolomics Profiles Identified by CE-TOFMS between High and Low Intramuscular Fat Amount in Fattening Pigs

**DOI:** 10.3390/metabo10080322

**Published:** 2020-08-07

**Authors:** Masaaki Taniguchi, Aisaku Arakawa, Motohide Nishio, Toshihiro Okamura, Chika Ohnishi, Kouen Kadowaki, Kimiko Kohira, Fumika Homma, Kazunori Matsumoto, Kazuo Ishii

**Affiliations:** 1Institute of Livestock and Grassland Science, National Agriculture and Food Research Organization, Tsukuba Ibaraki 305-0901, Japan; aisaku@affrc.go.jp (A.A.); mtnishio@affrc.go.jp (M.N.); okamut@affrc.go.jp (T.O.); kazishi@affrc.go.jp (K.I.); 2Miyazaki Station, National Livestock Breeding Center, Kobayashi Miyazaki 886-0004, Japan; c0ohnisi@nlbc.go.jp; 3Ibaraki Station, National Livestock Breeding Center, Chikusei Ibaraki 308-0112, Japan; k0kadowk@nlbc.go.jp; 4National Livestock Breeding Center, Nishigo Fukushima 961-8511, Japan; k0kohira@nlbc.go.jp (K.K.); f0honma@nlbc.go.jp (F.H.); k0matsumoto@nlbc.go.jp (K.M.)

**Keywords:** biomarker, CE-TOFMS, intramuscular fat, meat quality, metabolomics, porcine

## Abstract

The amount of intramuscular fat (IMF) present in the loin eye area is one of the most important characteristics of high-quality pork. IMF measurements are currently impractical without a labor-intensive process. Metabolomic profiling could be used as an IMF indicator to avoid this process; however, no studies have investigated their use during the fattening period of pigs. This study examined the metabolite profiles in the plasma of two groups of pigs derived from the same Duroc genetic line and fed the same diet. Five plasma samples were collected from each individual the day before slaughter. Capillary electrophoresis-time of flight mass spectrometry (CE-TOFMS) was used to analyze the purified plasma from each sample. Principle component analysis (PCA) and partial least squares (PLS) were used to find the semi-quantitative values of the compounds. The results indicate that branched-chain amino acids are significantly associated with high IMF content, while amino acids are associated with low IMF content. These differences were validated using the quantification analyses by high-performance liquid chromatograph, which supported our results. These results suggest that the concentration of branched-chain amino acids in plasma could be an indicative biomarker for the IMF content in the loin eye area.

## 1. Introduction

Pork consumers depend primarily on meat quality when making purchasing decisions. Intramuscular fat (IMF) content in the loin eye muscle, also known as marbling, is a particularly valuable trait of high-quality pork and is associated with the meat’s flavor. IMF content may vary by breed types, muscle types, genders, ages, feeding conditions, and final slaughter weight [1,2,3]. The heritability of IMF content in Duroc breed pigs has been estimated to be moderate to high (0.39 to 0.69) [4,5,6,7,8,9,10,11]. This has led to improvements in the phenotypic value of livestock in recent decades, despite that IMF is recognized as a polygenic trait [1,12].

Pigs grown on energy restricted diets accumulate significantly more IMF content when their gene expression profiles in relation to protein, glycogen and lipid turnover are altered [13]. Dietary regulation, particularly of lysine, has also resulted high amounts of IMF in pork [14]. A better understanding of the molecular and biological mechanisms underlying adipocytes development within muscle tissue could help to create effective nutritional controls during the fattening period of livestock, however few studies have been conducted to date.

Metabolomics are of particular interest in animal production as postmortem aging of meat can be predicted using time course analysis of capillary electrophoresis-time of flight mass spectrometer (CE-TOFMS) metabolomics. Such findings can contribute to the improvement of pork meat quality checks prior to slaughter [15]. More recently, Muroya et al. [16] proposed a novel concept, “MEATabolomics”, which combines the muscle biology and meat metabolomics of domestic animals. Additionally, metabolomics data can be defined as intermediate phenotypes, as metabolites are found between the genome level and the external phenotype level, such as growth rate, fat deposition and other economic traits. Metabolomics could lead to “next-generation phenotyping” approaches that improve the prediction of breeding values and selection schemes [17]. Furthermore, Suravajhala et al. [18] reported that high-throughput omics technologies have contributed to our understanding of complex biological phenomena, disease resistance, and holistic production improvements. Hence, metabolomics data may have a potential to improve prediction accuracy of genomic selection and to enhance the impact of breeding schemes for traits related to animal production by incorporating multiple layers of the high-throughput omics approach. In this context, development of non-invasive measurement technologies to obtain the metabolomics data is key for precise evaluation of target traits.

The objective of this study was to analyze and compare metabolite profiles in high and low IMF content pigs, in order to identify metabolites that could be used as IMF content indicators in pigs. To achieve this goal, we used a CE-TOFMS as our primary metabolomics technology due to its highly-sensitive, broad-range detection ability. We used the CE-TOFMS system to screen for candidate metabolites via a semi-quantitative method, then used an absolute quantification method to identify which candidate metabolites are associated with IMF content. We concluded that branched-chain amino acids (BCAA), glycine (Gly), anserine, and carnosine could be used as potential non-invasive biomarkers to estimate IMF content in the loin eye muscle of pigs before slaughter.

## 2. Results

### 2.1. Phenotypic Characteristics

Pigs were separated into low and high IMF groups at two locations of the National Livestock Breeding Center (NLBC), based on IMF (%) content in the loin eye muscle. The mean proportion of IMF content in pigs at the Miyazaki (MIY) station were 3.1% and 8.7%, respectively. The mean proportion of IMF content in pigs at the Ibaraki station (IBR) were 2.9% and 5.4%, respectively (Table 1). The differences were highly significant at both stations. Moisture (%) measured in the muscle tissues of low IMF pigs were significantly higher than in the high IMF pigs at both stations. At the MIY station, protein (%) and loin eye area (cm^2^) in low IMF pigs were significantly 8% and 11% higher than in high IMF pigs, respectively. At the IBR station, cooking loss (%) was significantly 15% higher than those of high IMF pigs.

### 2.2. Screening of Differential Metabolomics Profiles with CE-TOFMS

The total number of metabolites detected in the MIY samples were 201 (144 cations and 57 anions), which were used as the semi-quantitative values (relative area based on the internal standard). Likewise, the IBR metabolites detected were 152 (104 cations and 48 anions). The mean detection rates of metabolites, which are defined as the proportion of metabolites identified among either 201 (MIY) or 152 (IBR), were 82.0% and 88.6%, respectively. The mean detection rate of metabolites at IBR was greater than at MIY, despite the fact that more metabolites were detected at MIY than IBR.

Principal component analysis (PCA) was carried out based on the semi-quantitative values (relative area) of all detected metabolites. At MIY, the high IMF group tended to be in the PC1 positive and PC2 negative regions, while the low IMF group showed higher variability (Appendix A). The high IMF group at IBR was found in the PC3 and PC4 positive regions, but high and low IMF groups were neither separated by PC1 nor PC2 (Appendix A).

Partial least squares (PLS) was then used to identify possible relationships between IMF content and metabolite profiles at MIY and IBR (Figure 1. Appendix A). Since the PLS was performed between two groups (low or high IMF), we only considered PLS1 separation. All high IMF pigs and low IMF pigs were clearly in the positive and negative PLS1 scores. Therefore, we considered that positive loading values were associated with high IMF and negative loading values were associated with low IMF (Table 2). Statistically significant metabolites with positive loading values at MIY were leucine (Leu), *O*-acetylhomoserine/*2*-aminoadipic acid, 1-methylnicotinamide, choline, and phosphorylcholine, while all 10 metabolites detected with negative loading values were significant. Statistically significant metabolites with positive loading values at IBR were urea and gluconic acid, while those of the negative loading values were threonine (Thr) and diethanolamine. At both stations a BCAA, valine (Val) was identified from the positive loading metabolites. At MIY, Leu and isoleucine (Ile) were also identified from the positive loading metabolites. However, as we observed in the PCA analysis, BCAAs (Leu, Ile, and Val), tryptophan (Trp) and amino acid metabolites (*O*-acetylhomoserine/2-aminoadipic acid, *N,N*-dimethylglycine) were also detected in the list of positive PLS1 loading metabolites at MIY. Negative PLS1 loading metabolites at MIY included amino acids (Gly and beta-alanine (β-Ala)), amino acid metabolites (*N*-acetylornithine, *N*-acetyllisyne, 5-oxoproline, *N5*-ethylglutamine), peptides (anserine-divalent, carnosine) and organic acids (creatine, cis-aconitic acid). At IBR, urea, *3*-hydroxybuyric acid, nicotinamide, mucic acid and homocitrulline were commonly observed in the top positive loading metabolites on PCA3 and PCA4, While negative PLS1 loading metabolites consisted of several amino acids (Thr, Asn, Arg, Lys, methionine (Met), and Tyr) and amino acid metabolites (5-hydroxylysine and hydroxyproline).

### 2.3. Integrated Analysis of Metabolomics Data Using Absolute Quantification

With the CE-TOFMS metabolomics analysis system, the concentration in micro molar (mM) of 110 metabolites was calculated using a standard curve method. We then focused on the differences in these selectively quantified metabolites between the low and high IMF groups at the two stations. Consequently, the total number of metabolites detected in any of the eight samples from MIY and IBR was 59 mM. The mean number of cations detected at MIY was 38.7, and the mean number of anions at MIY was 34.6. At IBR, cations were 13.7, while anions were 11.9. The mean detection ratios of quantified metabolites at MIY were 88.8% and 78.9% at IBR. This indicates that there were a greater number of quantified metabolites identified at MIY than IBR. For the 59 quantified metabolites, we analyzed relationships between IMF content and the metabolites.

We used PLS to further determine which metabolites are definitively associated with IMF content. The PLS score plot showed that the low and high IMF groups at both stations were more separated by PLS2 than PLS1 or PLS3 (Figure 2), while the two stations themselves were clearly separated by PLS1 (Figure 2). We then extracted the top five metabolites with both positively and negatively high loading of PLS2 (Table 3, Appendix A). Positively loading PLS2 metabolites included the BCAAs Leu, Ile, and Val, and creatine, which were found to be significant metabolites with relatively moderate correlation coefficient (R). Comparison of these BCAAs and creatine concentrations at MIY indicated that the high IMF group displayed a significantly higher amount or tended to show greater amounts of the metabolites than those in the low IMF group. Basically, these same metabolites, BCAAs and creatine in the high IMF group at IBR also showed a greater amount than those of low IMF, although the difference was not as large as at MIY.

All metabolites from the negatively loading PLS2 metabolites were determined significant with slightly higher correlation coefficients as compared to those in the positively loading metabolites. Similar to the positive loading metabolites, the differences in the negatively loading PLS2 metabolites concentrations between the high and low IMF groups at MIY were clearer and more significant than those at IBR. Consequently, all metabolites identified by PLS2 factor loadings were commonly found in the list of metabolites which were identified with the one-factor PLS analysis at MIY and IBR (Table 2).

### 2.4. Comparison of Amino Acids and Related Metabolites

Since several amino acids and their related metabolites were identified as the differential metabolites between IMF groups, we next focused our analysis on those metabolites derived from BCAAs such as 3-methyl-2-oxovaleric acid (2K3MVA)/4-methyl-2-oxovaleric acid (2-oxoleucine), 2-oxoisovaleric acid (2-KIV) (Figure 3). The combined amount of Ile and Leu in the high IMF group at MIY was significantly higher than in the low IMF group. The amount of Ile and Leu at IBR, and Val at both stations showed a similar tendency, although the difference was not significant. Likewise, the amount of 2K3MVA/2-Oxoleucine and 2-KIV in the high IMF groups at both stations were greater than in the low IMF groups. There was no significant difference in the total free amino acid contents between the low and high IMF groups. The mean concentrations and standard deviations were 3428 ± 240 (μM) and 3258 ± 182 (μM), respectively.

### 2.5. Amino Acid Content Analysis in the Muscle Tissue of Pig Carcass

Per our CE-TOFMS metabolomics results, we concluded that BCAAs were associated with high IMF content in loin eye muscle. We examined the free amino acid contents of 20 amino acids with taurine, carnosine and anserine (Asn) in the muscle tissues of pig carcasses in order to better understand the mechanism by which high IMF content had either resulted from or was caused by high BCAAs in plasma (Table 4).

At MIY, high IMF pigs had significantly higher concentrations of Asn in loin eye muscle tissue than low IMF pigs, while Gly, carnosine and Asn in high IMF pigs were significantly lower than low IMF. The results of Gly and Carnosine matched the results shown in Table 2 and Table 3. At IBR, low IMF pigs had greater concentrations of Asp, His, Met, Trp, phenylalanine (Phe), Ile, and Leu in loin eye muscle tissues than those of high IMF pigs. The BCAA contents of loin eye muscles in low IMF pigs were significantly higher than those of high IMF pigs. This result is quite opposite from the results obtained from the plasma metabolomics. As seen in the plasma metabolomics, the total free amino acid content of loin eye muscle tissues displayed no difference between the low and high IMF groups.

## 3. Discussion

This study sought to identify the metabolites associated with IMF content in pork loin eye muscle using CE-TOFMS metabolomics data detected in pig plasma before slaughter. We selected five castrated boars with low IMF content, and five castrated boars with high IMF content in the loin eye muscle from two NLBC stations in Japan (IBR and MIY), for a total of 20 pigs. The difference in IMF content in the high IMF pigs was 2.8 times greater than the low IMF pigs at MIY. At IBR, the difference in IMF content in high IMF pigs was 1.9 times greater than the low IMF pigs. In contrast the moisture content (%), crude protein content (%), and loin eye area (cm^2^) in both high IMF groups were either significantly less or tended to be less than those of the low IMF groups. Most of the growth-related traits and the eating-quality traits displayed no differences between the groups. The metabolites differentially identified between the IMF content levels were therefore considered to be primarily associated with IMF content, and secondly with moisture content. Possible partial relationships may have occurred between the candidate metabolites for IMF content, crude protein content, loin eye area or cooking loss (Table 1).

Plasma samples from each pig were profiled using semi-quantification results derived from CE-TOFMS metabolomics. As indicated in the PCA analysis results, large variations were observed, particularly in the low IMF groups at both stations, although the high IMF groups had relatively close PCAs plots. No nutritional controls or genetic factors were used to generate these differences. Since no obvious separation was observed in the IMF content differences from either station, it can also be concluded that none of these factors were involved in samples obtained during this study and it is natural to see such vague separations (Appendix A).

Next, we conducted the PLS analysis. Our results clearly depicted variations in the candidate metabolites in the first PLS1 screening; more specifically, that high IMF groups had positive PLS1 values, while low IMF groups had negative PLS1 values (Figure 1). The metabolites with high positive loading values of PLS1 consistently included BCAAs and amino acid metabolites. This led us to conclude that BCAAs are potential biomarkers for IMF content in pork loin eye muscles. Metabolites with negative loading values of PLS1 included several kinds of amino acids and amino acid metabolites, which suggests that an increased amount of amino acids and amino acid metabolites in plasma may represent lower IMF content (i.e., leaner meat) (Table 2).

We used the quantification results from the CE-TOFMS to validate these findings. We applied the integrated PLS analysis using quantification data from the four groups (Figure 2). Although the PLS1 accounted for 60.4% of the total IMF variability, it appeared to obviously discriminate station differences, which was not the original purpose of this study. The PLS2 showed the comprehensible separation between the IMF content groups. Even though the number of metabolites quantified in the integrated analysis was 59 out of 201 at MIY and 152 at IBR during the screening analysis, BCAAs were commonly extracted from the list of metabolites with high loading PLS2 values (Table 3).

According to the correlation coefficients estimated in Table 2 and Table 3, metabolites with higher R values showed clear differences between the high and low IMF groups. Additionally, metabolite concentration differences between the two IMF groups at MIY were more obvious than those at IBR. Nevertheless, the concentration of metabolites in the high IMF groups was consistently greater at both stations, with the opposite holding true for metabolites with negative loading PLS2 values.

Since BCAAs were indicated to correspond to IMF content, we examined the potential BCAA-related metabolites closer (namely 2K3MVA, 2-oxoleucine, and 2-KIV) (Figure 3). Since 2K3MVA and 2-oxoleucine could not be separately identified by the CE-TOFMS system, additional amounts of 2K3MVA and 2-oxoleucine and their original BCAAs (Ile and Leu) were applied (Figure 3). The relative amount of metabolites degraded from BCAAs (2K3MVA, 2-oxoleucine, and 2-KIV) indicated a tendency similar to the original BCAAs. Even if the differences were not statistically significant, all of these metabolites follow the patterns of “low in low IMF”, and “high in high IMF”. One explanation for this may be that the baseline amount of BCAAs in high IMF pig plasma is greater than in low IMF pigs, which means that the amount of metabolized BCAAs might follow the same pattern. If this is the case, then it is quite important to analyze the amino acid content in tissues that metabolize amino acids.

Post slaughter amino acid analysis was conducted on the loin eye muscle tissues (Table 4). Results of this analysis were inconsistent with the metabolomic plasma analysis, that is, the amino acid tissue analysis showed that high IMF pigs at MIY had significantly lower amounts of Gly, Car, and Ans than low IMF pigs. This suggests that the amount of Gly, Car, and Ans (divalent) detected in the plasma of high IMF pigs might be correlated with the intake of those metabolites into muscle tissues. The amino acid analysis at IBR indicated that the amount of Asp and Met in the low IMF group were significantly greater than those in the high IMF group, which matched our plasma screening results. It is interesting that the amount of BCAA in the muscle tissues of the low IMF group were significantly greater than in the high IMF group. Additionally, the BCAA differences at MIY were not as obvious as in the plasma samples (N.S.), although the same trend was observed in both the plasma and the muscle tissues. These results suggest that the amount of BCAA in plasma is positively associated with high IMF content in the loin eye muscle.

Recently, it was demonstrated that surplus dietary Ile, one of BCAAs, increased the IMF content in skeletal muscle via the upregulation of lipogenic genes by stimulating lipogenesis in skeletal muscle tissue of finishing pigs [19]. In the previous study, Luo et al. clearly showed the effect of extra Ile in the diet on lipogenesis in the muscle tissue, serum cholesterol levels, and fatty acid composition without affecting growth performance. However, it is not possible to further discuss whether Ile (BCAA) intake into the muscle cells affected lipid synthesis due to the lack of Ile level in muscle tissue [19]. In contrast, it has been well known that BCAA are effective amino acids to regulate protein synthesis because of a greater increase in protein synthesis than degradation [20,21,22].

Therefore, one possible explanation for this is that the availability of BCAA and/or the intake of BCAA in the muscle tissue of low IMF pigs are more effective than in high IMF pigs, which results in the pig building more muscle mass. It is possible, therefore, that the availability of BCAA in the muscle tissue of high IMF pigs is relatively low, which would mean that some kind of molecular mechanism(s) could be triggered that could enhance the preadipocytes near the muscle cells.

The Duroc pigs used for this study were derived from the one previously established to improve IMF content at NLBC. The heritability of IMF in the Duroc genetic line established at NLBC has been previously determined to be 0.52 [23]. Although there were certain differences in the metabolite profiles between MIY and IBR by the PLS1 (Figure 2), these observed differences may be due to environmental factors only, since the genetic Duroc line and the feeding regimes were nearly identical at each location.

Previously, Katsumata et al. [14] reported that a low lysine diet fed to finishing pigs increased IMF content, although the authors recognized that the influence of restrictive amino acid nutrition on IMF content needs to be researched further. Restricted amino acid diets may increase stress sensitivity in pigs, since it has been indicated to do so in other animals [14,24,25]. In the case of beef marbling improvements, vitamin A restriction in the diet of beef steers increased IMF content due to hyperplasia of the adipocytes, although subcutaneous fat depth was not affected [26,27]. This method of feeding beef cattle diets with low vitamin A is well-known and used worldwide. In addition to the vitamin A restriction, it is suggested that a genotype of alcohol dehydrogenase 1C (ADH1C) has an interactive effect with vitamin A on IMF content, because of the differential transcriptional regulation that potentially occurs in this genotype [28]. It is plausible that nutritional control combined with certain genetic backgrounds may increase or decrease the IMF content in meat animals. However, there is still an inevitable phenotypic variation in IMF content, as is the nature of quantitative traits. Therefore, a non-invasive method, such as plasma metabolomics by CE-TOFMS, could be of benefit to determine the physiological status of meat producing animals, since it can measure a broader range of compounds than other metabolomics systems.

In conclusion, in this study we found that pigs with high IMF content in the loin eye muscle also had an increased amount of BCAA in their blood plasma. We also found that pigs with low IMF content in the loin eye muscle had an increased amount of Gly, Ans, and Car in their blood plasma. We suggest, therefore, that BCAA, Gly, Ans, and Car are potential biomarkers that could be used to estimate IMF in the loin eye muscle before slaughter. Further study of these biomarkers is required, since their amounts were not always consistently demonstrated in this study.

## 4. Materials and Methods

### 4.1. Animals

This study used castrated boars from the Duroc breed, which has been genetically bred for high IMF content as reported in the previous study [23]. All pigs used in this study were fed identical diets based on the Japan Nutrition Standards (NILGS 2016) and were raised at two locations of the NLBC (IBR and MIY) in Japan.

At each location, five high IMF content pigs and five low IMF content pigs were chosen from among the herd, for a total of 20 animals. Other than IMF content, growth performance and carcass characteristics of pigs were compared (Table 1). Aside from the fact that low IMF pigs took longer to gain slaughter weight, there were no significant phenotypic variables between any of the four IMF groups.

Blood samples were collected in a vacuum blood collection tube with EDTA/2Na from the jugular vein of each pig after fasting overnight. After blood samples were collected, the pigs were slaughtered at the abattoir located at each station. The blood collection tube was kept on ice and centrifuged with 3000× *g* at 4 °C for 10 min. Plasma was removed from the blood collection tube and placed into a new sample tube, then snap frozen, and stored in −80 °C freezer until use. All procedures involving animals were performed in accordance with the National Livestock Breeding Center’s guidelines for care and use of laboratory animals.

### 4.2. Semi-Quantitative Metabolomics (Basic Scan) by Capillary Electrophoresis-Time of Flight Mass Spectrometry (CE-TOFMS)

Preprocessing was initiated by adding 50 μL pig plasma to 450 μL methanol containing 50 μM Internal Standard Solution 1 (H3304-1002, Human Metabolome Technologies (HMT), Tsuruoka, Yamagata, Japan) followed by mixing. To the mixture, 500 μL chloroform and 200 μL ultrapure water were added and then mixed. After centrifugation at 2300× *g*, in 4 °C for 5 min, 400 μL supernatant of the mixture was removed to ultrafiltration using UltraFree MC PLHCC (HMT) and centrifuged at 9100× *g* at 4 °C for 120 min. The filtrate was once frozen-dried and dissolved in 50 μL ultrapure water just before applying to Agilent CE-TOFMS system (Agilent Technologies, Santa Clara, CA, USA). Metabolome analysis was performed by Basic Scan package of HMT using the CE-TOFMS based on the methods described previously [29,30]. Briefly, CE-TOFMS analysis was carried out using an Agilent CE capillary electrophoresis system equipped with an Agilent 6210 time-of-flight mass spectrometer (Agilent Technologies, Santa Clara, CA, USA). CE-TOFMS was operated using Agilent G2201AA ChemStation software version B.03.01 (Agilent Technologies, Santa Clara, CA, USA), compounds were detected using the cation or anion modes. Briefly, the compounds were electrophoresed through fused silica capillaries (50 μM × 80 cm, Agilent Technologies, Santa Clara, CA, USA) with 30 kV under 50 mbar and 10 s pressure injections. The electrophoresed compounds were ionized with either ESI Positive or Negative MS ionization. The scan range for the compounds was between 50 and 1000 *m*/*z*. Detected peaks from the CE-TOFMS system were processed by MasterHands ver2.17.1.11 [31]. Signal peaks corresponding to isotopomers, adduct ions, and other product ions of known metabolites were excluded, and remaining peaks were annotated according to the HMT metabolite database library, which includes 900 metabolites, based on their *m*/*z* values with the MTs determined by TOFMS. Areas of the annotated peaks were then normalized based on internal standard levels and sample amounts to obtain relative levels of each metabolite.

### 4.3. Absolute Quantification of 110 Target Metabolites for CE-TOFMS Analysis

Based on the same preprocessing and the analysis system, primary 110 metabolites involved in the pathways of glycolytic/gluconeogenesis, TCA cycle, pentose phosphate, lipid metabolites, and nucleic acid metabolites were absolutely quantified based on one-point calibrations using their respective standard compounds.

### 4.4. Measurement of Free Amino Acids, Peptides, and Carcass Traits in Muscle Tissues

The procedure used to analyze the free amino acids and peptides was carried out using the exact same method described in [32]. Briefly, 0.10 g samples of loin eye muscle tissues were drawn from minced raw meat (about 15 g in total). Samples were homogenized with 4.24 mL of ultrapure water, 4 mL of N-hexane and 0.16 mL of internal standard solution mixed with norvaline (5 nmol/μL) in ultrapure water, and then centrifuged at 1750× *g* for 5 min. The underlayer was mixed with 4 mL of N-hexane and centrifuged at 1750× *g* for 5 min. The resultant underlayer was then mixed with 3.6 mL of acetonitrile and centrifuged at 1750× *g* for 10 min. The resulting supernatant was filtered through a 0.45-μm microfilter (Millex-LH; Merck Millipore, Billerica, MA, USA), and the filtrate was then mixed with 45% acetonitrile solution and analyzed for free amino acids and peptides using an Agilent 1260 infinity high performance liquid chromatograph equipped with an Agilent 1260 Binary Pump, 1260 HiP Degasser, 1260 HiP ALS autosampler, 1290 thermostat, 1260 Thermostatted column compartment control module, 1260 diode array detector and a Poroshell 120 EC-C18 column (3.0 × 100 mm, 2.7 μm; Agilent). The eluents used were: (i) 20 mmol/L disodium hydrogen phosphate (pH 7.6); and (ii) acetonitrile/methanol/water (5:5:1, *v*/*v*/*v*). Amino acids and peptides were identified through the comparison of their retention times with those of established standards. The concentrations of each were calculated using internal and external standard solutions and expressed as μmol per g of meat and μmol per 1% of moisture in meat, respectively. The internal standard solutions were used to account for matter lost during analysis, and the external standard solutions (1, 10, 50, and 100 pmol/μL) were used to plot a calibration curve for each amino acid and peptide. Carcass traits were analyzed according to the methods described previously [32,33]. Moisture content was determined in duplicates by drying 2 g samples of meat drawn from the minced raw meat (approximately 30 g in total) for 24 h at 105 °C. IMF content was determined by Soxhlet extraction of the dried samples with diethyl ether for 16 h. Crude protein content was determined using 1 g samples of meat drawn from the minced raw meat (approximately 30 g in total) by the Kjeldahl method using a nitrogen distillation titration device (2400 Kjeltec auto sampler system, FOSS, Hillerod, Denmark). Cooking loss was determined by the weight difference of meat samples (approximately 50 g) before and after heating at 70 °C for 1 h in a water bath. After cooking loss measurement, the meat sample was then used for WBSF measurement with crosshead speed of 200 mm/min using Instron (model 5542; Instron, University Avenue Norwood, MA, USA). WHC was determined using meat samples (approximately 0.5 g) drawn from the minced raw meat which were centrifuged by 10,000× *g* for 30 min. Determination of tenderness, pliability, toughness, and brittleness were performed using meat samples prepared for the same as the WBSF using Tensipresser (model TTP-50BX II, Takemoto Electric Inc., Tokyo, Japan).

### 4.5. Data Analysis

Metabolomics data acquired by the CE-TOFMS system were normalized and the semi-quantitative and absolute quantification data were then analyzed by PCA and PLS [34] using MATLAB (The MathWorks, Natick, MA, USA) and R programs [35] developed by HMT Co. Ltd. Differences between high and low IMF groups were examined by Welch’s *t*-test. Significant difference was defined by *p* < 0.05.

## Figures and Tables

**Figure 1 metabolites-10-00322-f001:**
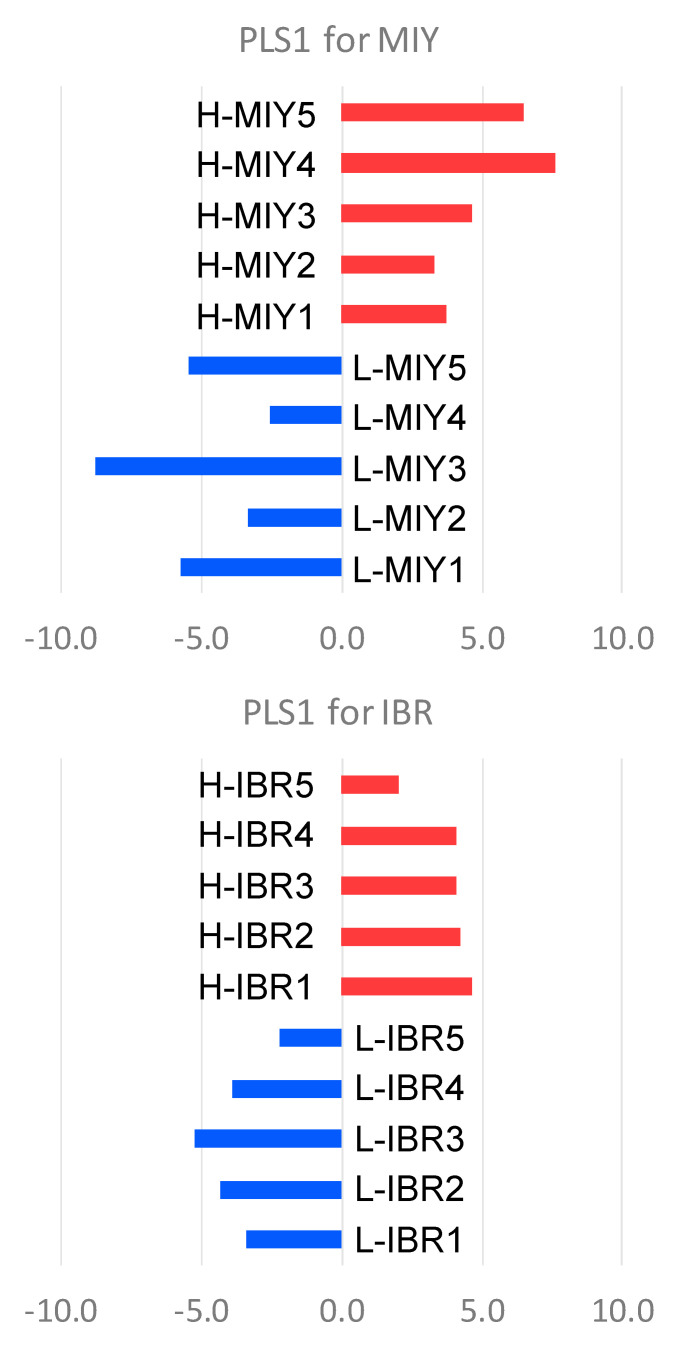
Partial least squares (PLS) analysis by the metabolites detected from pig plasma samples showing the difference in the intramuscular fat contents at the two stations. Blue and red bars indicate low and high IMF pigs, respectively. Top and bottom panes show the results of PLS in MIY and IBR stations, respectively.

**Figure 2 metabolites-10-00322-f002:**
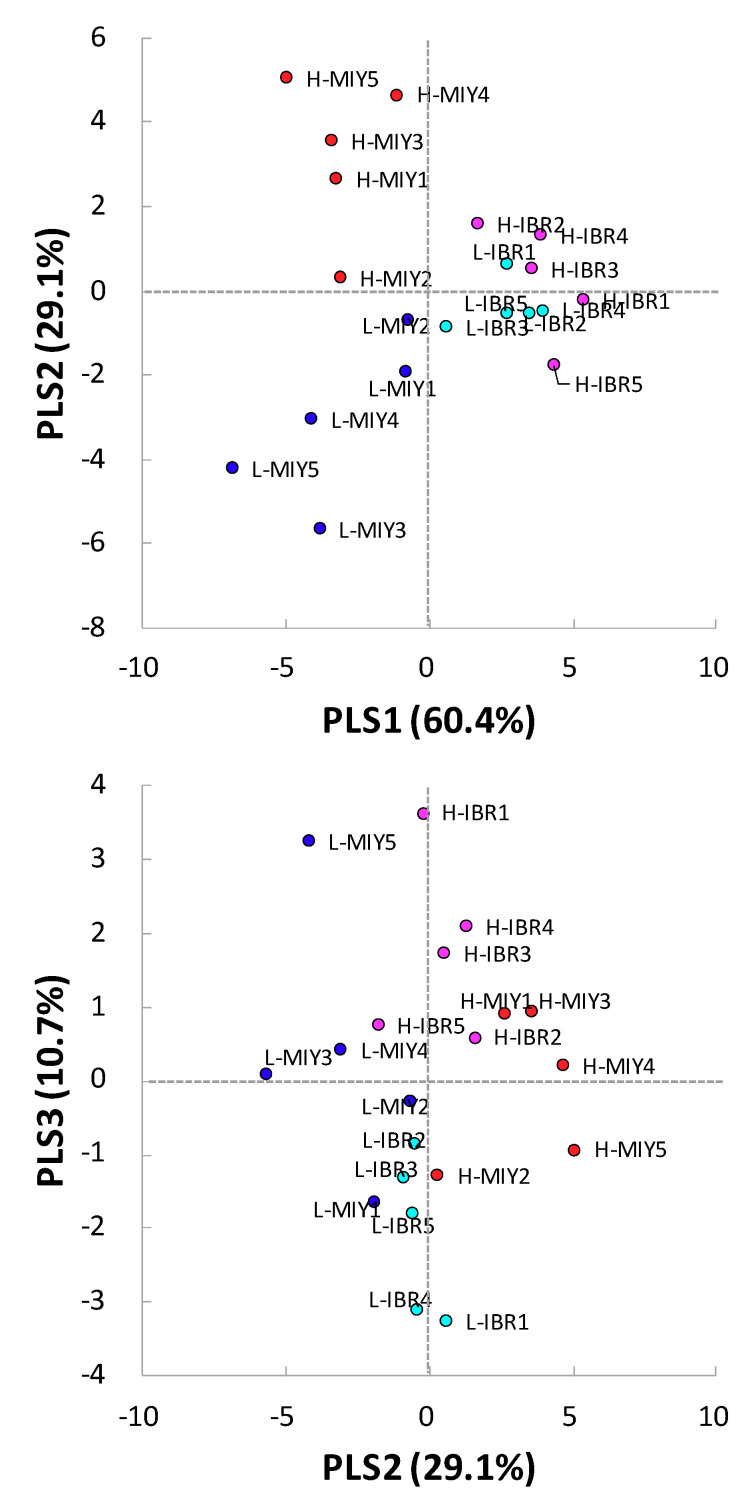
Integrated analyses of partial least squares by the metabolites detected from pig plasma samples showing the difference in the intramuscular fat contents at the two stations. Blue and red dots indicate low and high IMF pigs from MIY station, respectively. Cyan and magenta dots indicate low and high IMF pigs from IBR station, respectively. Top and bottom panes show the results of PLS analyses with PLS1 and 2 and with PLS2 and 3, respectively.

**Figure 3 metabolites-10-00322-f003:**
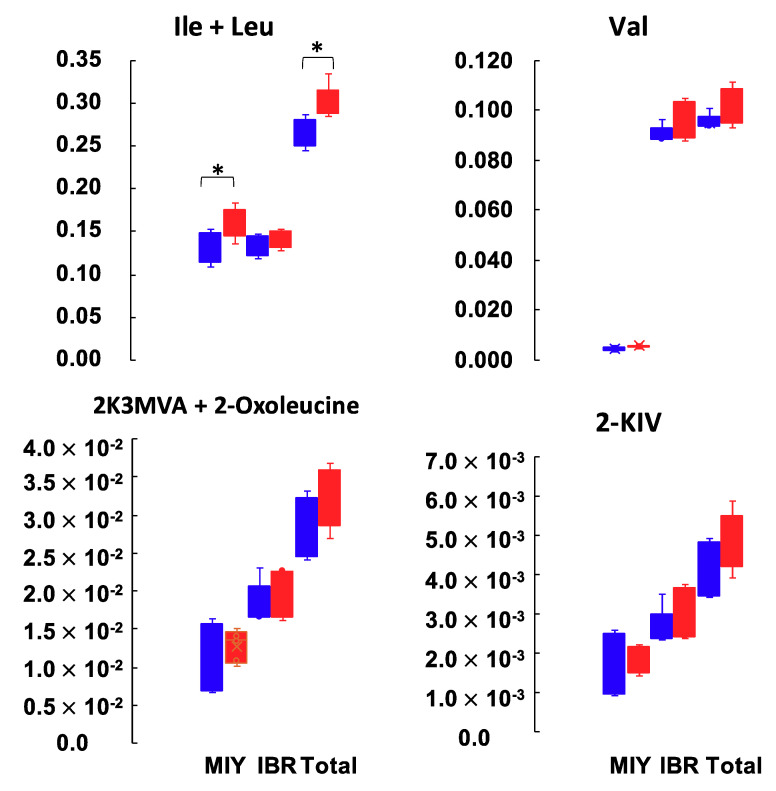
Comparisons of branched-chain amino acids and their degraded metabolites detected from pig plasma samples showing the difference in the intramuscular fat contents at the two stations. Blue and red bars indicate low and high IMF pigs, respectively. Total shows the sum of MIY and IBR. *Y*-axis indicate the relative area of each metabolites normalized by the internal standard. 3-methyl-2-oxovaleric acid (2K3MVA) and 4-methyl-2-oxovaleric acid (2-oxoleucine), which are the degraded metabolites of Ile and Leu, respectively, could not be separately detected by the capillary electrophoresis-time of flight mass spectrometer (CE-TOFMS). Statistical significance is defined as * (p < 0.05).

**Table 1 metabolites-10-00322-t001:** Comparison of growth performance and carcass characteristics between high and low intramuscular fat (IMF) groups at two stations.

Growth Performance and Carcass Characteristics	Miyazaki (MIY) Station			Ibaraki (IBR) Station		
	L-IMF MIY	H-IMF MIY	*p* *^1^	L-IMF IBR	H-IMF IBR	*p* *^1^
IMF (%)	3.101 ± 0.255	8.699 ± 1.441	*p* < 0.001	2.868 ± 0.550	5.447 ± 0.797	*p* < 0.001
Days old at slaughter	149.2 ± 9.99	152.2 ± 13.64	*N.S.*	154.4 ± 8.357	169.8 ± 6.997	*p* < 0.05
Moisture (%)	73.96 ± 0.339	69.98 ± 0.966	*p* < 0.001	73.81 ± 0.978	71.48 ± 0.444	*p* < 0.01
Crude protein (%)	21.82 ± 0.173	20.23 ± 0.917	*p* < 0.01	22.35 ± 0.149	22.02 ± 0.350	*N.S.*
Cooking loss (%)	24.68 ± 1.548	26.52 ± 1.823	*N.S.*	28.04 ± 1.966	24.38 ± 2.269	*p* < 0.05
WHC (%) *^2^	68.30 ± 5.489	72.30 ± 4.511	*N.S.*	68.14 ± 5.824	68.94 ± 4.199	*N.S.*
WBSF (kg) *^3^	2.392 ± 0.338	2.048 ± 0.489	*N.S.*	2.963 ± 0.550	2.838 ± 0.478	*N.S.*
Tenderness (kg/cm^2^)	38.09 ± 5.586	34.81 ± 2.010	*N.S.*	46.45 ± 6.442	44.82 ± 6.710	*N.S.*
Pliability	1.497 ± 0.0918	1.375 ± 0.0609	*N.S.*	1.527 ± 0.0442	1.494 ± 0.0679	*N.S.*
Toughness (kg/cm^2^ × cm)	8.163 ± 1.474	8.054 ± 1.082	*N.S.*	10.27 ± 1.900	10.15 ± 1.888	*N.S.*
Brittleness	1.631 ± 0.0750	1.662 ± 0.0864	*N.S.*	1.568 ± 0.0880	1.632 ± 0.110	*N.S.*
Average daily gain (g/day)	1141.3 ± 62.70	1035.9 ± 188.8	*N.S.*	1006.7 ± 53.91	1008.8 ± 46.73	*N.S.*
Loin eye area (cm^2^)	35.06 ± 1.318	31.45 ± 2.059	*p* < 0.05	35.10 ± 3.369	34.29 ± 3.219	*N.S.*
Back fat thickness (cm)	2.597 ± 0.547	3.376 ± 0.548	*N.S.*	2.044 ± 0.568	2.476 ± 0.661	*N.S.*
Statue height (cm)	63.16 ± 1.895	61.22 ± 0.722	*N.S.*	65.48 ± 1.001	64.16 ± 2.330	*N.S.*
Body length (cm)	107.6 ± 3.855	105.28 ± 2.285	*N.S.*	89.7 ± 39.35	106.5 ± 5.040	*N.S.*

*^1^ Statistical significance is defined when *p* < 0.05. *N.S.* denotes no significance (*p* > 0.05). *^2^ WHC: Water-holding capacity (%) by centrifugation at 10,000× *g*. *^3^ WBSF: Warner-Bratzler Shear Force value (Kg) measured after cooling of the carcass.

**Table 2 metabolites-10-00322-t002:** Top 10 metabolites with positive and negative high loading values in partial least square at two stations.

MIY Station			IBR Station		
Metabolites	PLS1		Metabolites	PLS1	
	*R* *^1^	*p*		*R* *^1^	*p*
Positive loading					
Leu	0.731	1.6 × 10^−2^	Urea	0.729	1.7 × 10^−2^
*O*-acetylhomoserine 2-aminoadipic acid	0.727	1.7 × 10^−2^	Gluconic acid	0.654	4.0 × 10^−2^
1-Methylnicotinamide	0.711	2.1 × 10^−2^	3-hydroxybutyric acid	0.544	1.0 × 10^−1^
Choline	0.645	4.4 × 10^−2^	Isethionic acid	0.505	1.4 × 10^−1^
Phosphorylcholine	0.637	4.8 × 10^−2^	Nicotinamide	0.494	1.5 × 10^−1^
Ile	0.609	6.1 × 10^−2^	Val	0.493	1.5 × 10^−1^
Creatine	0.603	6.5 × 10^−2^	Taurine	0.488	1.5 × 10^−1^
Val	0.582	7.8 × 10^−2^	Mucic acid	0.476	1.6 × 10^−1^
*N,N*-dimethylglycine	0.569	8.6 × 10^−2^	Homocitrulline	0.475	1.7 × 10^−1^
Trp	0.470	1.7 × 10^−1^	Sarcosine	0.452	1.9 × 10^−1^
Negative loading					
Gly	−0.861	1.4 × 10^−3^	Thr	−0.716	2.0 × 10^−2^
Anserine_divalent	−0.840	2.3 × 10^−3^	Diethanolamine	−0.705	2.3 × 10^−2^
*N*-acetylornithine	−0.834	2.7 × 10^−3^	Thymidine	−0.611	6.0 × 10^−2^
*N*-acetyllysine	−0.796	5.9 × 10^−3^	5-hydroxylysine	−0.589	7.3 × 10^−2^
5-oxoproline	−0.753	1.2 × 10^−2^	Asn	−0.566	8.8 × 10^−2^
Carnosine	−0.741	1.4 × 10^−2^	Arg	−0.533	1.1 × 10^−1^
Creatinine	−0.707	2.2 × 10^−2^	Lys	−0.521	1.2 × 10^−1^
*N^5^*-ethylglutamine	−0.705	2.3 × 10^−2^	Hydroxyproline	−0.514	1.3 × 10^−1^
cis-Aconitic acid	−0.685	2.9 × 10^−2^	Met	−0.505	1.4 × 10^−1^
β-Ala	−0.683	2.9 × 10^−2^	Tyr	−0.459	1.8 × 10^−1^

*^1^
*R* indicates correlation coefficient between PLS score and each metabolite levels. Since *O*-acetylhomoserine/2-aminoadipic acid were detected within the identical single peaks having the consistent *m*/*z* (molecular mass/electric charge of ions) and MT/RT (migration time/retention time) by the CE-TOFMS system employed in this study, the values in the table indicated the combination of two compounds: The *m*/*z* and MT/RT for *O*-acetylhomoserine/2-aminoadipic acid is 162.076 and 12.10, respectively.

**Table 3 metabolites-10-00322-t003:** Top 5 metabolites with positive and negative high loading values of PLS2 in the integrated analysis regarding IMF content at two stations.

Metabolites	*R* *^1^	*p* *^2^	MIY Station			IBR Station			Total		
			Low IMF	High IMF	*p* *^2^	Low IMF	High IMF	*p* *^2^	Low IMF	High IMF	*p* *^2^
**Positively Loading**											
Leu	0.610	***	153.33 ± 6.18	180.6 ± 6.56	*	156.41 ± 6.46	165.16 ± 6.13	*N.S.*	154.86 ± 5.81	172.88 ± 7.00	*
Ile	0.577	***	93.43 ± 9.25	120.37 ± 8.14	*tnd.*	107.85 ± 4.29	110.69 ± 2.94	*N.S.*	100.64 ± 7.65	115.53 ± 6.20	*
Val	0.531	*	236.99 ± 19.06	284.70 ± 13.93	*tnd.*	268.24 ± 4.32	285.19 ± 9.64	*N.S.*	252.62 ± 14.94	284.94 ± 11.29	*
Creatine	0.529	*	232.13 ± 34.19	307.37 ± 8.54	*tnd.*	302.38 ± 4.64	317.32 ± 10.72	*N.S.*	267.25 ± 28.34	312.34 ± 9.44	*
Trp	0.370	*tnd.*	60.06 ± 6.08	71.31 ± 4.34	*N.S.*	74.12 ± 2.92	72.079 ± 3.35	*N.S.*	67.08 ± 5.59	71.69 ± 3.66	*N.S.*
**Negatively Loading**											
Gly	−0.788	***	920.72 ± 48.07	666.51 ± 22.31	**	829.52 ± 14.96	799.15 ± 43.97	*N.S.*	875.12 ± 39.86	732.83 ± 45.36	***
Creatinine	−0.611	***	102.50 ± 5.45	82.42 ± 4.56	*	89.64 ± 3.82	89.99 ± 3.83	*N.S.*	96.07 ± 5.38	86.20 ± 4.35	*tnd.*
Hydroxyproline	−0.573	***	81.09 ± 11.06	51.67 ± 7.05	*tnd.*	66.47 ± 3.25	55.19 ± 5.81	*N.S.*	73.78 ± 8.42	53.43 ± 6.15	*
β-Ala	−0.506	*	6.362 ± 0.755	4.164 ± 0.347	**	6.388 ± 0.323	6.522 ± 0.793	*N.S.*	6.375 ± 0.547	5.343 ± 0.801	*N.S.*
Carnosine	−0.443	*tnd.*	10.97 ± 0.590	8.239 ± 0.646	*	11.52 ± 1.03	11.80 ± 1.01	*N.S.*	11.245 ± 0.800	10.019 ± 1.157	*N.S.*

The unit of metabolites concentration is represented in μM. *^1^
*R* indicates correlation coefficient between PLS score and each metabolite levels. *^2^ Values are denoted with means ± S.E.M. Statistical significance is defined as *tnd.* (*p* < 0.10), * (*p* < 0.05), ** (*p* < 0.01), and *** (*p* < 0.001). *N.S.* indicates no significance.

**Table 4 metabolites-10-00322-t004:** Analysis of amino acid contents in the loin eye muscle tissues from two stations.

Amino Acid *^1^	MIY Station			IBR Station		
	Low IMF	High IMF	*p* *^2^	Low IMF	High IMF	*p* *^2^
Asp	9.6 × 10^−2^ ± 8.1 × 10^−3^	9.5 × 10^−2^ ± 2.1 × 10^−2^	*N.S.*	8.1 × 10^−2^ ± 3.3 × 10^−2^	3.2 × 10^−2^ ± 1.0 × 10^−2^	*
Glu	5.5 × 10^−1^ ± 1.0 × 10^−1^	6.2 × 10^−1^ ± 1.2 × 10^−1^	*N.S.*	6.8 × 10^−1^ ± 1.5 × 10^−1^	5.7 × 10^−1^ ± 9.3 × 10^−2^	*N.S.*
Asn	1.6 × 10^−1^ ± 2.8 × 10^−2^	2.3 × 10^−1^ ± 5.0 × 10^−2^	*	2.6 × 10^−1^ ± 6.9 × 10^−2^	2.4 × 10^−1^ ± 1.7 × 10^−2^	*N.S.*
Ser	4.9 × 10^−1^ ± 9.1 × 10^−2^	4.8 × 10^−1^ ± 9.9 × 10^−2^	*N.S.*	5.5 × 10^−1^ ± 1.2 × 10^−1^	4.5 × 10^−1^ ± 5.1 × 10^−2^	*N.S.*
Gln	1.658 ± 0.250	1.490 ± 0.144	*N.S.*	1.485 ± 0.438	9.9 × 10^−1^ ± 1.4 × 10^−1^	*tnd.*
His	1.7 × 10^−1^ ± 2.0 × 10^−2^	1.7 × 10^−1^ ± 2.6 × 10^−2^	*N.S.*	1.9 × 10^−1^ ± 2.3 × 10^−2^	1.5 × 10^−1^ ± 2.3 × 10^−2^	*
Gly	1.62 ± 0.26	1.213 ± 0.0966	*	1.852 ± 0.498	1.789 ± 0.481	*N.S.*
Thr	3.4 × 10^−1^ ± 5.9 × 10^−2^	3.6 × 10^−1^ ± 6.2 × 10^−2^	*N.S.*	3.9 × 10^−1^ ± 7.5 × 10^−2^	3.3 × 10^−1^ ± 3.1 × 10^−2^	*N.S.*
β-Ala	6.7 × 10^−1^ ± 2.3 × 10^−1^	4.4 × 10^−1^ ± 1.0 × 10^−1^	*N.S.*	5.1 × 10^−1^ ± 8.3 × 10^−2^	5.6 × 10^−1^ ± 1.6 × 10^−1^	*N.S.*
Arg	3.6 × 10^−1^ ± 5.9 × 10^−2^	3.5 × 10^−1^ ± 6.6 × 10^−2^	*N.S.*	4.0 × 10^−1^ ± 8.3 × 10^−2^	2.9 × 10^−1^ ± 3.5 × 10^−2^	*tnd.*
Ala	2.143 ± 0.325	2.329 ± 0.397	*N.S.*	2.643 ± 0.472	2.170 ± 0.293	*N.S.*
Tau	2.341 ± 0.441	2.16 ± 0.289	*N.S.*	2.404 ± 0.772	1.829 ± 0.920	*N.S.*
Car	26.41 ±1.56	20.99 ± 2.35	**	28.95 ± 1.65	27.60 ± 2.20	*N.S.*
Ans	8.5 × 10^−1^ ± 6.0 × 10^−2^	6.6 × 10^−1^ ± 8.4 × 10^−2^	**	7.1 × 10^−1^ ± 5.7 × 10^−2^	7.3 × 10^−1^ ± 1.2 × 10^−1^	*N.S.*
Tyr	2.8 × 10^−1^ ± 4.7 × 10^−2^	2.7 × 10^−1^ ± 4.8 × 10^−2^	*N.S.*	3.0 × 10^−1^ ± 5.8 × 10^−2^	2.5 × 10^−1^ ± 2.3 × 10^−2^	*N.S.*
Val	3.9 × 10^−1^ ± 6.1 × 10^−2^	4.8 × 10^−1^ ± 5.7 × 10^−2^	*tnd.*	5.3 × 10^−1^ ± 8.0 × 10^−2^	4.4 × 10^−1^ ± 5.6 × 10^−2^	*N.S.*
Met	2.5 × 10^−1^ ± 4.1 × 10^−2^	2.7 × 10^−1^ ± 4.6 × 10^−2^	*N.S.*	3.0 × 10^−1^ ± 3.4 × 10^−2^	2.3 × 10^−1^ ± 2.1 × 10^−2^	**
Trp	8.8 × 10^−2^ ± 7.8 × 10^−3^	9.9 × 10^−2^ ± 5.9 × 10^−3^	*tnd.*	1.2 × 10^−1^ ± 9.8 × 10^−3^	1.0 × 10^−1^ ± 5.2 × 10^−3^	**
Phe	3.3 × 10^−1^ ± 4.4 × 10^−2^	3.5 × 10^−1^ ± 4.1 × 10^−2^	*N.S.*	3.9 × 10^−1^ ± 4.0 × 10^−2^	3.4 × 10^−1^ ± 1.2 × 10^−2^	*
Ile	3.1 × 10^−1^ ± 5.7 × 10^−2^	3.4 × 10^−1^ ± 5.1 × 10^−2^	*N.S.*	3.8 × 10^−1^ ± 4.9 × 10^−2^	3.1 × 10^−1^ ± 2.5 × 10^−2^	*
Leu	7.0 × 10^−1^ ± 1.2 × 10^−1^	6.3 × 10^−1^ ± 1.0 × 10^−2^	*N.S.*	7.3 × 10^−1^ ± 1.1 × 10^−1^	5.5 × 10^−1^ ± 6.2 × 10^−2^	*
Lys	4.4 × 10^−1^ ± 8.1 × 10^−2^	4.2 × 10^−1^ ± 8.1 × 10^−2^	*N.S.*	4.5 × 10^−1^ ± 8.1 × 10^−2^	3.5 × 10^−1^ ± 3.5 × 10^−2^	*tnd.*
Hyp	3.142 ± 0.253	2.897 ± 1.630	*N.S.*	2.099 ± 0.198	2.054 ± 0.0751	*N.S.*
Pro	9.6 × 10^−1^ ± 8.9 × 10^−2^	2.790 ± 3.31	*N.S.*	5.4 × 10^−1^ ± 1.04 × 10^−1^	3.7 × 10^−1^ ± 1.3 × 10^−1^	*tnd.*
FAA *^3^	14.42 ± 0.732	15.67 ± 3.60	*N.S.*	15.26 ± 2.76	12.40 ± 1.87	*N.S.*

The unit of amino acid concentration is represented in μmol/g. *^1^ This list includes analysis result of amino acids together with Taurine, Carnosine and Anserine. *^2^ Statistical significance is defined as *tnd.* (*p* < 0.10), * (*p* < 0.05), ** (*p* < 0.01). *N.S.* indicates no significance. *^3^ FAA denotes total amount of free amino acids.

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
