# Peer review of "Differential Metabolomics Profiles Identified by CE-TOFMS between High and Low Intramuscular Fat Amount in Fattening Pigs"

_metabolites, 2020, doi:10.3390/metabo10080322_

Round 1

Reviewer 1 Report

Comments to the authors

Major comments:

The manuscript from Taniguchi et al. clearly showed that plasma BCAA concentration and IMF content in the loin eye area, suggesting they might be an indicative biomarker for the IMF content in pork meat. The reviewer suggests accepting this manuscript after the authors response following comments. I hope these comments listed below will be helpful for the authors.

1. Is there such relationship between plasma BCAA concentration and IMF content in the loin eye area in other porcine strain?

2. The authors need to discuss about relationship between BCAA metabolism and IMF content.

3. Blood relationship among the 20 pigs used in this study should be explained.

Minor comment:

1. Line 67. CE-TOFMS is once defined (Line 51).

2. Line 324. 4 Cº → 4 Cº

3. Line 325. -80 Cº → -80 Cº

4. Please describe methods for parameters shown in Table 1 (e.g., moisture,
crude protein, cooking loss, and so on).

Author Response

RE: Revision of Manuscript–Metabolites-862126

To Reviewer 1,

On behalf of all authors, I would like to thank you for reviewing our manuscript entitled “Differential Metabolomics Profiles Identified by CE-TOFMS Between High and Low Intramuscular Fat Amount in Fattening Pigs” manuscript ID: Metablites-862126.

According to your comments and suggestions, we modified our manuscript to add informative content and improve the readability.

Please see the below answers and comments [blue letters] to you.

We believe that we could do our best to improve the contents of our study.

Sincerely yours,

Masaaki Taniguchi, Ph. D., corresponding author.

July 2020

Reviewer 2 Report

The material and methods chapter should be located before the results chapter.

The discussion chapter is largely a description of obtained the results.

There is little discussion with the results of other researchers.

However, this could be due to innovation of the study.

Author Response

RE: Revision of Manuscript–Metabolites-862126

To Reviewer 2,

On behalf of all authors, I would like to thank you for reviewing our manuscript entitled “Differential Metabolomics Profiles Identified by CE-TOFMS Between High and Low Intramuscular Fat Amount in Fattening Pigs” manuscript ID: Metablites-862126.

According to your comments and suggestions, we modified our manuscript to add informative content and improve the readability.

Please see the below answers and comments [blue letters] to you.

We believe that we could do our best to improve the contents of our study.

Sincerely yours,

Masaaki Taniguchi, Ph. D., corresponding author.

July 2020
